# Determining the Characteristics of Papers That Garner the Most Significant Impact: A Deep Dive into Mexican Engineering Publications

**Claudia N. Gonzalez Brambila** [1,*] **, José I. Ponce** [1]**, Silvia B. Gonzalez Brambila** [2] **and Matias F. Milia** [1]

1    Department of Business Administration, Instituto Tecnológico Autónomo de México—ITAM, Mexico City 01080, Mexico; jose.ponce@itam.mx (J.I.P.); matias.milia@itam.mx (M.F.M.)
2    Department of Systems, Universidad Autónoma Metropolitana-Azcapotzalco—UAM-A, Mexico City 02200, Mexico; sgb@azc.uam.mx
*    Correspondence: cgonzalez@itam.mx; Tel.: +52-55-5628-4000

**Abstract:** Engineers make things, make things work, and make things work better and easier. This kind of knowledge is crucial for innovation, and much of the explicit knowledge developed by engineers is embodied in scientific publications. In this paper, we analyze the evolution of publications and citations in engineering in a middle-income country such as Mexico. Using a database of all Mexican publications in Web of Science from 2004 to 2017, we explore the characteristics of publications that tend to have the greatest impact; this is the highest number of citations. Among the variables studied are the type of collaboration (no collaboration, domestic, bilateral, or multilateral), the number of coauthors and countries, controlling for a coauthor from the USA, and the affiliation institution of the Mexican author(s). Our results emphasize the overall importance of joint international efforts and suggest that publications with the highest number of citations are those with multinational collaboration (coauthors from three or more countries) and when one of the coauthors is from the USA. Another interesting result is that single-authored papers have had a higher impact than those written through domestic collaboration.

**Keywords:** research impact; engineering; Mexico; article citation analysis; collaboration





## 1. Introduction

Knowledge and ideas are becoming more important aspects of economic competitiveness than assets and resources [1]. According to the World Bank, the four pillars of the knowledge economy are: education and training, information infrastructure, economic incentives and institutional regimes, and innovation systems [2]. More than ever, the knowledge embodied in human capital and in technology is central to economic development, and knowledge-based economies show higher rates of economic growth and competitiveness in all economic sectors [3]. As a result, knowledge production, transmission, and transfer are critical aspects for promoting growth, development, and increasing welfare.

Furthermore, to address current and future global challenges, the collective knowledge of researchers, institutions, and countries is required to achieve breakthroughs. Studies such as [4–6] Coccia and Wang [4], Chen, Zhang, and Fu [5], and Kwiek [6] have found that increased collaboration is associated with better quality and impactful knowledge. Other studies [7,8] have shown the immense growth of scientific collaboration, mainly in science and engineering. Among the multiple benefits of collaboration are sharing knowledge and expertise, tackling more complex problems, fertilizing ideas, and making better use of the scientific infrastructure [9].

Engineers make things, make things work, and make things work better and easier. They also use their creativity to design and implement systems, processes, and solutions that benefit mankind. Engineering disciplines integrate scientific principles with practically

oriented research [10] to provide innovative solutions to a wide range of industries. Much of the explicit knowledge developed by engineers is embodied in scientific publications, and the number of citations in those publications is increasingly seen as an indicator of the impact and quality of the knowledge embodied in those publications [11]. However, to the best of our knowledge, little is known about the characteristics of Mexican engineering publications that have the greatest impact.

Analyzing the impact and characteristics of engineering publications, especially in a country like Mexico, is essential because engineering plays a critical role in supporting the growth and development of the country's economy as well as in improving the quality of life.

Our study makes three contributions to the literature. First, it analyzes the evolution of knowledge creation in engineering in an advanced developing country, Mexico. Second, it explores the impact of collaboration on the number of citations a paper receives. Third, it studies the characteristics of the papers that tend to receive the highest number of citations. For the first contribution, knowledge creation from 2004 to 2017 in all types of engineering is presented. The method of measuring research collaboration is based on the coauthorship of papers. We distinguish among four types of collaboration [12]: no collaboration [solo-authored papers], domestic collaboration, bilateral collaboration, and multilateral collaboration. Domestic collaborations are papers written exclusively by researchers affiliated with a Mexican institution. Bilateral collaboration is used for those papers whose coauthors are affiliated with a Mexican institution and other institutions from another country. Multilateral collaboration involves researchers from Mexico and at least two other countries. Finally, for the third contribution, other articles' characteristics that are analyzed are the number of coauthors, the number of countries, a coauthor from the USA, and affiliation to the most productive Mexican institutions, which are used as control variables.

The article is structured as follows: the next section presents a literature review. The third section describes the data and models. The fourth section shows the results. Finally, the fifth section presents the conclusions.

## 2. Literature Review

Since the seminal work of Zuckerman and Merton [13], several studies have shown that contemporary science is increasingly collaborative [4,14]. Among the benefits that scientific collaboration brings [9,15] are: complementary knowledge and expertise to tackle the increasing complexity of problems; cross-fertilization of ideas to create knowledge or technology; access to a wide variety of resources; and a decrease in the costs of collaboration due to the advancement of information and communication technologies. Thus, there is multiple evidence of the rising numbers of coauthored papers and that multi-author papers receive more citations than solo-authored research. Moreover, international collaboration is associated with the greatest impact.

In a study of team size in chemistry from 1910 to 1960, de Solla Price [16], forecasts that, considering the trend of collaboration, by 1980, zero percent of the papers would be created by solo authors. Adams et al. [17] explore trends in the size of scientific teams and in institutional collaborations in elite American research universities from 1981 to 1999. They find that not only team size increased by 50%, but also the geographical dispersion was larger. They also find evidence that team size is positively related to output and influence.

Exploring almost 20 million papers plus 2 million patents, Wuchty et al. [7] find that in sciences, engineering, and social sciences, there has been a steady growth in the number of publications, and team size increased from 1.9 to 3.5 authors per paper over 45 years, starting in 1955. Moreover, they found that in science and engineering, teamwork has increased in 99.4% of the 171 subfields. Related to the impact, measured by the number of citations, teams produced more highly cited work in all broad areas of research than solo authors.

Examining the Italian production of Web of Science (WoS) from 2004 to 2010, Abramo and D'Angelo [18] confirm that, in almost all subject categories, there is a consistent and linear growth in the citability of a publication with the number of co-authors. They also find that the correlation between citations and authors varies depending on the document type. For conference proceedings, the correlation is weaker compared to articles, and in engineering reviews, the increase is even larger than for articles.

Hsiehchen et al. [19] analyzed four decades of publications in WoS, covering natural, social, and applied research disciplines. They found that the number of authors and countries has steadily risen, and the proportion of single-authored papers has dropped fast over time. Moreover, they calculated that the probability of not being cited has decreased and the probability of being highly cited has increased in collaborative multinational papers compared to one-nation papers.

Coccia and Wang [4] find that although international scientific collaboration has increased in volume in all research fields, in engineering and technology, the level of internationally coauthored papers has been lower than in other more basic fields such as astronomy or physics.

In a study that covers observations from 1900 to 2011, Larivière et al. [20] confirm that an increase in the number of authors, addresses, or countries leads to an increase in impact. However, diminishing citation returns have resulted from the constant inflation of collaboration since the beginning of the last century, so larger teams are necessary to realize a higher impact.

In an analysis that covers publications in 1995 and 1996 and citations in a three-year window for 50 selected countries, Glänzel [21] confirms that international collaboration has intensified in the last decade and that, on average, these publications get higher citation rates than domestic publications. Specifically, he shows that for all fields of knowledge, the share of international co-publications in Mexico changed from 29.9% to 42.6% between 1985/86 and 1995/96. Related to the citation impact, he finds that the relative expected citation index of international co-publications in 1995/96 for engineering is 0.10.

## 3. Materials and Methods

### 3.1. Data

This paper explores the characteristics of Mexican engineering publications that tend to receive the highest number of citations.

The study is based on all articles registered from 2004 to 2017 in the eighteen categories classified as engineering in the WoS Core Collection. We collapsed the eighteen categories into seven broad categories, like LOC [22], for the purpose of analysis. The classification is shown in Table 1.

**Table 1.** Engineering categories.

| Mechanics | Civil | Electronics | Chemistry | Management | Geotechnics | Biologics | Others |
|---|---|---|---|---|---|---|---|
| Aerospace | Civil | Computer science software | Agricultural | Industrial | Geological | Biomedical | Multidisciplinary |
| Mechanical | | Electrical & electronic | Chemical | Manufacturing | Metallurgy & metallurgical | Cell & tissue | |
| | | | Environmental | | Petroleum | | |
| | | | Marine | | | | |
| | | | Ocean | | | | |

Source: Own elaboration with information from LOC [22].

It is important to point out that not all Mexican engineering research products are published in WoS. Other outcomes not considered in this paper include patents, books, proceedings, consulting reports, research projects, prototypes, startups, and articles in journals not included in WoS. Still, one of the main advantages of using WoS publications

is that it is an objective measure that covers the whole country in all engineering fields, and other studies have and could use the same source and reproduce the results to compare them with other countries or research areas [23].

Bibliometric data were collected from the WoS between March and April 2022. A specific query was written to capture publications related to engineering categories for the period 1970 to 2021:

CU=Mexico AND (SU=AEROSPACE OR SU=AGRICULTURAL OR SU=BIOMEDICAL OR SU=CELL & TISSUE OR SU=CHEMICAL OR SU=CIVIL OR SU=COMPUTER SCIENCE SOFTWARE OR SU=ELECTRICAL & ELECTRONIC OR SU=ENVIRONMENTAL OR SU=GEOLOGICAL OR SU=INDUSTRIAL OR SU=MANUFACTURING OR SU=MARINE OR SU=MECHANICAL OR SU=METALLURGY & METALLURGICAL OR SU=MULTIDISCIPLINARY OR SU=OCEAN OR SU=PETROLEUM)

The data contained 46,722 publications indexed in WoS, published by at least one Mexican author from 1970 to 2021. However, we decided to focus our attention only on articles published from 2004 to 2017. This is 13,322 articles and 135,927 citations. The decision was based first on the fact that the articles have received many more citations (mean 19.02 and standard deviation 31.50) than other types of publications. For example, the mean number of citations for proceedings is 0.8085 with a standard deviation of 0.0214; the mean number of citations for books is 0.4766; and the mean number of citations for other types of publications is 0.27. Second, we focus on articles from 2004 to 2017 because the number of articles started to grow significantly since 2004. According to the Science and Technology Indicators produced by the National Science Foundation (https://ncses.nsf.gov/pubs/nsb20214/data, accessed on 16 August 2023), the average rate of global publication output in engineering from 1997 to 2003 was 0.05%, and in 2004 and 2005 it was 18% and 23%, respectively. Scopus was created in 2004. The growth rate between 1970 and 2003 was much lower (1.9%) than after 2004, which has been more than 3 times larger (6.6%). We restricted the analysis to 2017 because we consider the number of citations in a 5-year window, considering that a larger proportion of citations are received in the first 5 years of publication; this is the year of publication and the next 4 years [24]. We also exclude all articles with nine or more authors, considering that these kinds of publications have different characteristics of collaboration [25]. Figure 1 shows the evolution of publications per year and the number of citations per year.

As seen in Table 2, there are significant differences in the number of publications and citations among different types of engineering. It is important to stress that the purpose of this analysis is to highlight differences, as there are more broad areas of knowledge [26]. This does not mean that researchers are more or less productive, depending on the field of engineering. Moreover, 58% of the papers are classified in more than one type of engineering; given this overlap, the sum of the papers by type of engineering surpasses the total number of all papers in engineering.

Electronics is the type of engineering that congregates the highest number of publications, and the article with the highest number of citations in the sample is in this area. It is important to highlight that some articles in this area are also included in other fields such as chemistry and biologics, for example, those publications related to the design of instruments or equipment for pharmaceutical purposes. Thus, this area includes multidisciplinary articles.

The field with the highest average number of citations is biologics, which is also the field with the highest average number of coauthors, countries, and proportion of international collaboration, either bilateral or multilateral. On the contrary, civil engineering has the smallest number of publications, the average number of citations, and the least proportion of international collaboration.

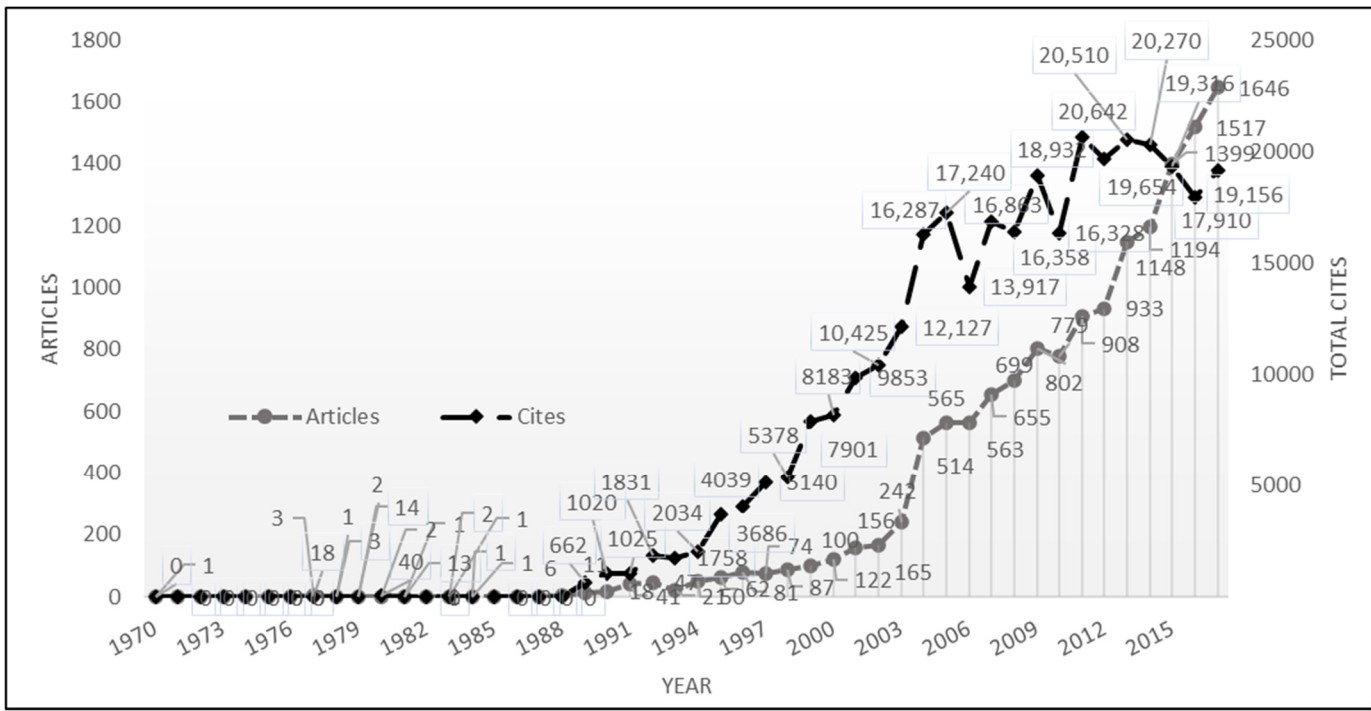

**Figure 1.** Evolution of Mexican publications and total citations in Engineering in WoS, 2004–2017. Source: Own elaboration based on WoS data.

**Table 2.** Descriptive statistics by type of engineering, 2004–2017.

| Type of Engineering | Articles | Mean Total Cites | Std. Dev. | Mean # of Coauthors | Mean # of Countries |
|---|---|---|---|---|---|
| All engineering | 13,322 | 19.02 | 31.50 | 4.08 | 1.39 |
| Biologics | 2294 | 23.87 | 38.52 | 4.42 | 1.47 |
| Chemistry | 4120 | 17.47 | 26.15 | 4.11 | 1.40 |
| Civil | 426 | 7.00 | 8.34 | 3.72 | 1.34 |
| Electronics | 8156 | 20.48 | 32.51 | 4.25 | 1.38 |
| Geotechnics | 2628 | 17.42 | 24.98 | 4.00 | 1.37 |
| Management | 702 | 16.15 | 22.96 | 3.56 | 1.31 |
| Mechanics | 1227 | 16.20 | 25.20 | 3.53 | 1.37 |
| Others | 1448 | 11.98 | 17.77 | 4.08 | 1.44 |

Source: Own elaboration based on WoS data. Note: Articles may be classified into more than one category. The total number of articles is 13,322.

### 3.1.1. Coauthorship

Figure 2 shows that over the period of analysis, there has been a steady growth (24%) in the average number of coauthors in all engineering research areas. This increasing trend in coauthorship is similar to what Thelwall and Maflahi [27] found. However, as was seen in Table 2, there are differences in the size of teams among the different types of engineering. The largest teams are in biologics followed by electronics, and the smallest are in mechanics and management. Related to the mean number of countries, the differences are quite small, so measuring this indicator by type of collaboration seems more appropriate.

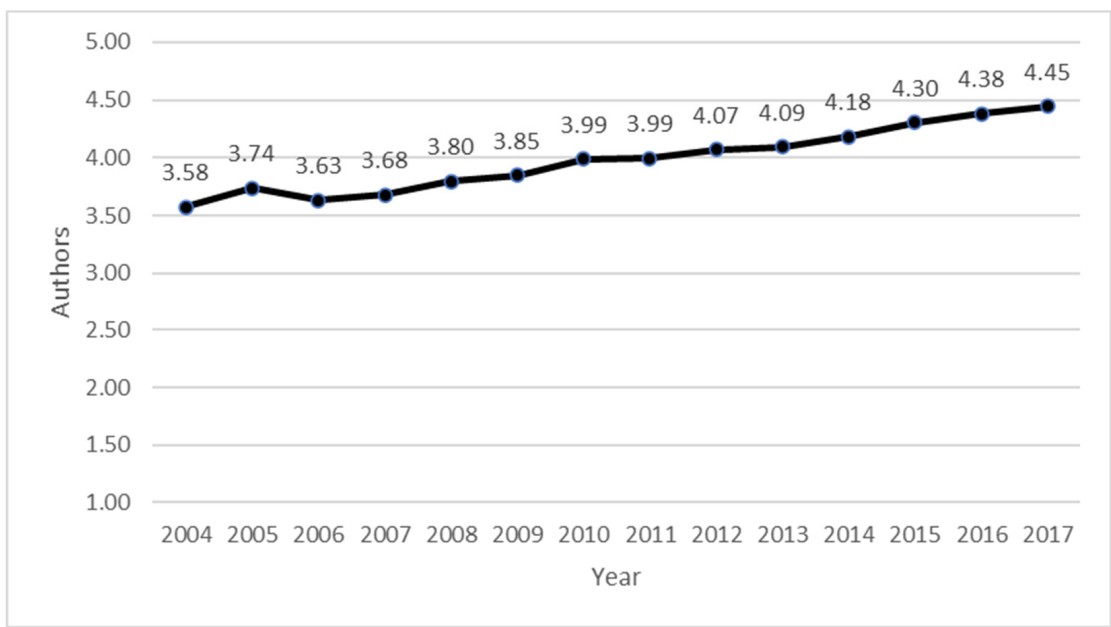

**Figure 2.** Evolution of the mean of coauthors, 2004–2017. Source: Own elaboration based on WoS data.

### 3.1.2. Type of Collaboration

Figure 3 shows that most of the knowledge that is produced in engineering in Mexico involves collaboration, mainly international (56%), either bilateral or multilateral, and only 3% of the articles are solo-authored papers. Domestic collaboration is the largest form, but it is the one that has grown the least (158%); on the other side, multilateral collaboration used to be the smallest, but it is the one that has grown the most (511%) and already exceeded bilateral collaboration, which over the period of analysis grew 204%.

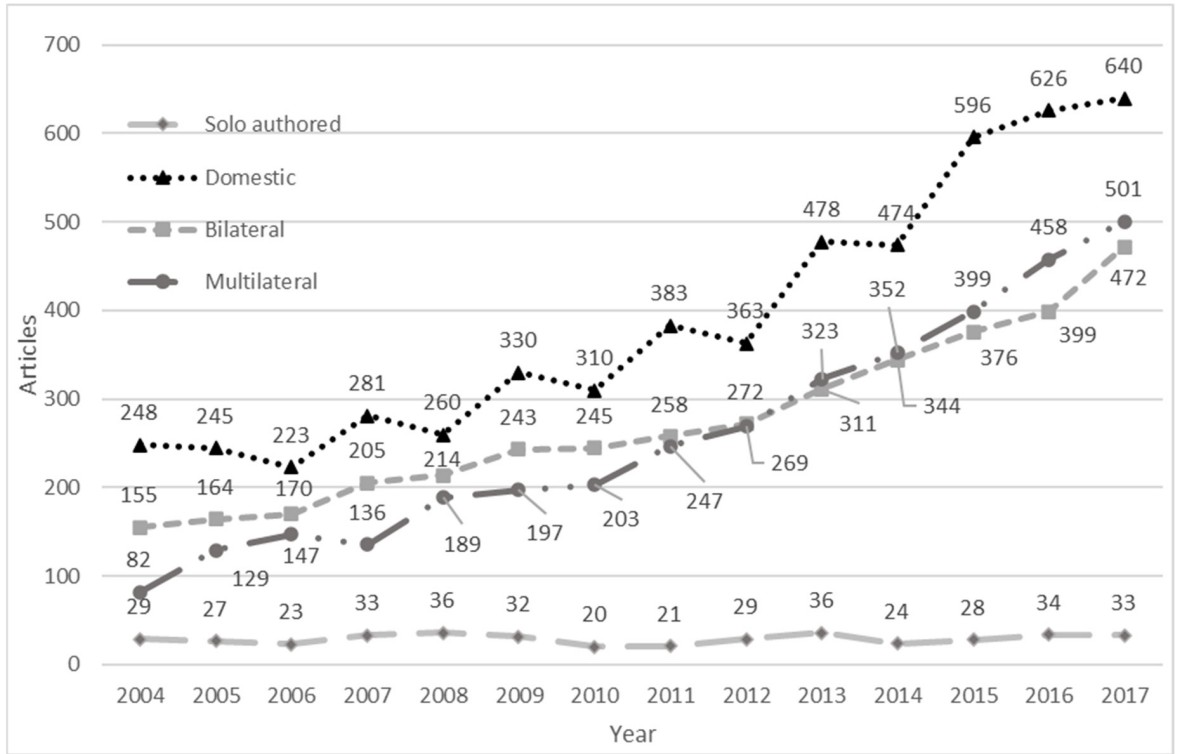

**Figure 3.** Evolution of engineering articles by type of collaboration, 2004–2017. Source: Own elaboration based on WoS data.

Many studies have shown that not all types of collaboration produce the same impact [20,28]. Table 3 shows that domestic collaboration is the type of collaboration that receives the least number of citations. Surprisingly, on average, solo-authored papers received even more citations than papers written under this type of collaboration. This contrasts with Wuchty et al. [7], findings that suggest that teams produce exceptionally high-impact research in comparison with solo works.

**Table 3.** Descriptive statistics by type of collaboration, 2004–2017.

| Type of Collaboration | Obs | Mean of Total Cites | Std. Dev. |
|---|---|---|---|
| Total | 13,322 | 19.02 | 31.50 |
| Solo authored | 405 | 17.70 | 36.23 |
| Domestic | 5457 | 15.08 | 24.83 |
| Bilateral | 3828 | 20.16 | 32.66 |
| Multilateral | 3632 | 23.92 | 37.42 |

Source: Own elaboration based on WoS data.

More than half (56%) of all the knowledge that has been created in engineering in Mexico involves international collaboration, and this is the form of collaboration that, on average, has the highest impact, mainly multilateral collaboration.

Table 4 shows the total number of publications by type of engineering and type of collaboration. As can be seen, electronics is the type of engineering with the highest collaboration; only 2.11% of the knowledge production is solo-authored papers. Biologics is the field with the highest international collaboration, either bilateral or multilateral (71%). As was highlighted before, civil engineering has the smallest proportion of international collaboration.

**Table 4.** Articles by type of engineering, proportion by type of collaboration, 2004–2017.

| Type of Engineering | Obs | % Solo Authored | % Domestic | % Bilateral | % Multilateral |
|---|---|---|---|---|---|
| All engineering | 13,322 * | 3.19% | 37.61% | 31.67% | 27.53% |
| Biologics | 2294 | 2.22% | 26.94% | 37.66% | 33.17% |
| Chemistry | 4120 | 3.96% | 30.53% | 37.60% | 27.91% |
| Civil | 426 | 4.23% | 51.64% | 20.42% | 23.71% |
| Electronics | 8156 | 2.11% | 43.45% | 27.51% | 26.92% |
| Geotechnics | 2628 | 4.30% | 33.83% | 35.96% | 25.91% |
| Management | 702 | 4.84% | 44.16% | 31.48% | 19.52% |
| Mechanics | 1227 | 4.07% | 44.82% | 26.16% | 24.94% |
| Others | 1448 | 4.70% | 35.22% | 29.07% | 31.01% |

* A total of 58% of the articles are in more than one engineering category. Source: Own elaboration based on WoS data.

### 3.2. Model

To study the impact of a paper, it is assumed that the baseline function is:

$$Y_1 = f(X_i, C_i) \tag{1}$$

Two different proxies of a paper's impact were considered. As in other papers such as Ruano-Ravina and Álvarez-Dardet [29] and Guo et al. [30], the total number of citations a paper has received is the first dependent variable. Considering that older papers have received more citations just because they have been published for more years, the number of citations a paper has received in the first five years of publication was the second dependent

variable [24]. $X_i$ are the independent and control variables (a detailed description of each variable can be found in Appendix A):

- Variables related to the team of coauthors:
  - Number of coauthors;
  - Number of countries.
- Variables related to the type of collaboration:
  - Solo authored (Solo);
  - Domestic collaboration (Dom);
  - Bilateral collaboration (Bi);
  - Multilateral collaboration (Multi).
- Control variables:
  - At least one coauthor from the USA.
- Variables related to the most productive institutions in Mexico:
  - UNAM (Universidad Nacional Autónoma de Mexico);
  - IPN (Instituto Politécnico Nacional);
  - UAM (Universidad Autónoma Metropolitana);
  - CINVESTAV (Centro de Investigación y de Estudios Avanzados del IPN);
  - IMP (Instituto Mexicano del Petróleo);
  - UDG (Universidad de Guadalajara);
  - UGU (Universidad de Guanajuato);
  - Other institution.
- $C_i$ is the error term.

The reason for including a coauthor from the USA as a control variable was because 48.9% of the papers with international collaboration have at least one coauthor from the USA. Narvaez-Berthelemot et al. [31] also find that the main international collaborator of Mexican researchers is the USA. Controlling for the seven most productive universities as institutions of affiliation was necessary because there is a wide dispersion in the size and productivity of Mexican universities [32].

Considering the nature of the data, a negative binomial (NB) model is used. This model is used when the dependent variable takes integer values and the variance is significantly greater than the mean. Moreover, NB models relate the dependent variable Y to one or more predictor variables, $X_i$, which can be quantitative or categorical. The procedure fits a weighted least squares model. Likelihood ratio tests were performed to test the significance of the model coefficients. Table 5 shows the descriptive statistics of the variables used.

**Table 5.** Descriptive statistics of all variables.

| Variable | Obs | Mean | Std. Dev. | Min | Max |
|---|---|---|---|---|---|
| Total Cites | 13,322 | 19.02 | 31.50 | 0 | 779 |
| Cites five-year window | 13,322 | 8.82 | 12.45 | 0 | 243 |
| # of Coauthors | 13,322 | 4.08 | 1.61 | 1 | 8 |
| # of Countries | 13,322 | 1.39 | 1.02 | 0 | 7 |
| Solo | 13,322 | 0.03 | 0.17 | 0 | 1 |
| Dom | 13,322 | 0.41 | 0.49 | 0 | 1 |
| Bi | 13,322 | 0.29 | 0.45 | 0 | 1 |
| Multi | 13,322 | 0.27 | 0.44 | 0 | 1 |
| USA | 13,322 | 0.49 | 0.50 | 0 | 1 |

Source: Own elaboration.

Table 6 shows the correlation among the nine variables in our models. As expected, there is a high correlation between multilateral collaboration and the number of countries, as well as between papers written with at least one USA coauthor and domestic and multilateral collaboration. Thus, none of the models include highly correlated variables. Moreover, to prove that our model has no autocorrelation problems, we submitted it to the serial correlation test of Wooldridge [33]; the results confirm that our models do not have such a problem.

**Table 6.** Correlation matrix.

| | Total Cites | Cites Five-Year Window | # of Coauthors | # of Countries | Solo | Dom | Bi | Multi | USA |
|---|---|---|---|---|---|---|---|---|---|
| Total cites | 1 | | | | | | | | |
| Cites five-year window | 0.7358 | 1 | | | | | | | |
| # of Coauthors | 0.0176 | 0.0804 | 1 | | | | | | |
| # of Countries | 0.1082 | 0.1704 | 0.2664 | 1 | | | | | |
| Solo | −0.0074 | −0.0204 | −0.3378 | −0.1031 | 1 | | | | |
| Dom | −0.1042 | −0.1394 | −0.1131 | −0.4851 | −0.1473 | 1 | | | |
| Bi | 0.023 | 0.0218 | −0.0431 | −0.2604 | −0.112 | −0.5274 | 1 | | |
| Multi | 0.0944 | 0.1413 | 0.3116 | 0.8355 | −0.1075 | −0.5061 | −0.3848 | 1 | |
| USA | 0.0955 | 0.1385 | 0.3143 | 0.4198 | −0.1733 | −0.8082 | 0.401 | 0.5596 | 1 |

Source: Own elaboration based on WoS data.

Different specification models were considered to analyze how the effect of one independent variable is moderated when other variables are included. Thus, five different models were run.

- Model 1: Number of coauthors, controlling for coauthors from the USA;
- Model 2: Number of countries, controlling for coauthors from the USA;
- Model 3: Number of coauthors and number of countries, controlling for coauthors from the USA;
- Model 4: Variables related to the type of collaboration;
- Model 5: Number of coauthors, variables related to the type of collaboration, controlling for the most productive institutions in Mexico.

Alternative models, like models 1 to 4, were also run for the most productive institutions in Mexico. The results are similar to those discussed in the next section.

## 4. Results

Table 7 shows the results of the regressions when the total number of citations is used as a dependent variable, and Table 8 shows the results when the number of citations in a five-year window is used. As can be seen, the results are similar for both dependent variables in all specification models. Thus, there is also one discussion for both tables.

### 4.1. Number of Coauthors

The number of coauthors has a positive and significant effect in models 1 and 5. However, in model 3, when the number of countries is included, it loses significance, suggesting that the number of countries captures most of the effect. For an impactful paper, the number of countries is more critical than the number of co-authors. The results of the NB model confirm the evidence of other research [18], where the greater the number of coauthors, the greater the impact on the citations received by the article. However, our results are in contrast to those of Puuska et al. [34], who find that when the effect of the number of authors is included, the citation impact between international and domestic

collaboration is minimal. In the Mexican case, the effect of the number of authors does not significantly reduce the size of the coefficients of the type of collaboration.

**Table 7.** Regression results from models 1 to 5. The dependent variable is the total number of citations.

| | Total Citations (1) | Total Citations (2) | Total Citations (3) | Total Citations (4) | Total Citations (5) |
|---|---|---|---|---|---|
| # Author | 0.0176 *** (0.0064) | | 0.0051 (0.0063) | | 0.0174 *** (0.0067) |
| # Countries | | 0.2036 *** (0.0154) | 0.2018 *** (0.0156) | | |
| Solo | | | | 0.0862 (0.0566) | 0.1533 ** (0.0603) |
| Bilat | | | | 0.3102 *** (0.0235) | 0.3312 ** (0.0235) |
| Multi | | | | 0.5184 *** (0.0237) | 0.5116 *** (0.0254) |
| USA | 0.3475 *** (0.0204) | 0.2444 *** (0.0210) | 0.2405 *** (0.0215) | | |

Standard error in parentheses. ** Significant at 0.5%. *** Significant at 0.1%. All models have time effects. Model 5 has institutional effects. Source: Own elaboration based on the negative binomial regression model results.

**Table 8.** Regression results from models 1 to 5. The dependent variable is cited in a five-year window.

| | Citations 5 Yrs (1) | Citations 5 Yrs (2) | Citations 5 Yrs (3) | Citations 5 Yrs (4) | Citations 5 Yrs (5) |
|---|---|---|---|---|---|
| # Author | 0.0185 *** (0.0063) | | 0.0053 (0.0063) | | 0.0192 *** (0.0067) |
| # Countries | | 0.2076 *** (0.0152) | 0.2057 *** (0.0154) | | |
| Solo | | | | 0.1400 ** (0.0570) | 0.1998 *** (0.0613) |
| Bilat | | | | 0.3009 *** (0.0232) | 0.3182 *** (0.0234) |
| Multi | | | | 0.5245 *** (0.0235) | 0.5123 *** (0.0252) |
| USA | 0.3515 *** (0.0202) | 0.2441 *** (0.0209) | 0.2041 *** (0.0214) | | |

Standard error in parentheses. ** Significant at 0.5%. *** Significant at 0.1%. All models have time effects. Model 5 has institutional effects. Source: Own elaboration based on the negative binomial regression model results.

### 4.2. Number of Countries

The effect of the number of countries on the number of citations is positive and significant in all models, suggesting that the diversification of countries increases the impact of research. These results confirm what Larivière et al. [20] have found, confirming that an increase in the number of countries produces a greater impact.

### 4.3. Type of Collaboration

The results suggest that publications written by a solo author or under binational or multinational collaboration tend to receive more citations than papers written under domestic collaboration. However, the highly cited articles are those written under multilateral collaboration (three or more countries). Our results coincide with those of Li and Li [12], who also find that multilateral collaboration, followed by bilateral collaboration, produces the most impactful knowledge. A surprising result is that articles written under domestic collaboration [only Mexican coauthors collaborate] receive fewer citations than single-authored papers. This result must be taken with caution considering the small number of publications with solo authors since the variance for these articles is relatively large.

### 4.4. Coauthor from the USA

Our results suggest that having a coauthor from the USA has a positive and significant impact on all models. The effect of collaborating with at least one USA coauthor on the number of citations received is positive and significant in all models. The results agree with what is presented by Sud and Thelwall [35] and Chinchilla-Rodríguez et al. [11], in the sense that coauthors whose institution is in countries such as the USA or some countries of Europe can get more citations than in other regions of the planet.

The results of the models that include control of the most productive universities do not change significantly; this suggests that the size of the institution is not a critical impact driver.

## 5. Conclusions

This paper shows the large increase in knowledge created by Mexican engineers in the last decades. Even though the mean number of coauthors per paper has not increased a lot, multilateral collaboration is the form of collaboration that has grown the most. Withstanding this, domestic collaboration is still the largest form of collaboration. The analysis also shows the great differences among the different types of engineering.

The results of the NB model reveal the characteristics of the most impactful articles in engineering, either for the total number of citations or citations in a five-year window. Confirming the results of other studies, Larivière et al. [20], Abramo and D'Angelo [18], and Li and Li [12], international collaboration produces more impactful knowledge, especially multilateral collaboration. The size of the coefficient suggests that multilateral collaboration increases the number of citations more than the number of coauthors, countries, or any other type of collaboration. Surprisingly, the results also reveal that solo-authored articles receive more citations than articles written only by Mexican coauthors; this is a domestic collaboration. This result is the opposite of what was found by Abramo and D'Angelo [18] and Lariviére et al. [20] in the sense that the greater the number of coauthors, the higher the impact, including solo-authored papers. The other visible result is that articles written in collaboration with a researcher from the USA significantly increase the impact of the paper.

As stated before, there are important limitations when the analysis of the creation of knowledge and its impact only considers articles and citations in WoS. In all areas of knowledge, and especially in engineering, there are many other outputs and outcomes, such as patents, human resources, linkages with firms, and consulting, among others, that this study is not considering. Thus, we are aware that there is an underestimation of the engineering research developed in Mexico. As in many other bibliometric studies, this one does not consider the qualitative aspects of collaboration. As stated by Katz and Martin [9], many other products of collaboration do not end in publication. Moreover, this study does not consider the environments and reward incentives that the Mexican government and institutions have created to encourage publications and collaboration, as well as other forms of knowledge creation.

In future research, it could be interesting to investigate how the diversity of teams, in terms of the personal characteristics of the coauthors, such as area of knowledge, gender, and age, affect the production and impact of the knowledge created. Other studies, such as Dasgupta, Scircle, and Hunsinger [36]; Huyer [37]; and Cheryan et al. [38], have documented the relatively low proportion of women in engineering and the gaps in productivity between women and men [24] in other countries. Thus, it would be interesting to explore this aspect in the Mexican context. Another relevant aspect is a deeper analysis of the specific characteristics of the networks and performing a similar analysis using other impact indicators, such as those introduced by Kanellos et al. [39].

Even though there are critical limitations to this study, we believe that our findings provide important insights for the design of policies that could encourage collaboration to produce more impactful knowledge to find better solutions to the increasingly complex and multidisciplinary problems that our planet is facing, in which engineering plays a critical role.

**Author Contributions:** Conceptualization, C.N.G.B. and J.I.P.; methodology, C.N.G.B. and J.I.P.; software, J.I.P. and C.N.G.B.; writing—original draft preparation, C.N.G.B., J.I.P., S.B.G.B. and M.F.M.; writing—review and editing, C.N.G.B., J.I.P., S.B.G.B. and M.F.M.; supervision, C.N.G.B. All authors have read and agreed to the published version of the manuscript.

**Funding:** This research was funded by Consejo Nacional de Ciencia y Tecnología (CONACyT-México) within the framework of the CONACYT project number A1S9013, "Evaluation of the impact of public policies on scientific, technological and innovative productivity in Mexico".

**Data Availability Statement:** The sources for the datasets are publicly available on Web of Science. The derived datasets presented in this study are openly available in the repository OpenICPSR at: https://doi.org/10.3886/E191261V1 (accesed on 2 December 2021).

**Acknowledgments:** Support from the Asociación Mexicana de Cultura, A.C., is appreciated. We also thank Daniel Rubí and Yamil Sanchez for data collection.

**Conflicts of Interest:** The authors declare no conflict of interest.

## Appendix A

**Table A1.** Variables used in the negative binomial regression model.

| Variable | Description |
| --- | --- |
| **Collaboration** | |
| Coauthors | Indicates the number of coauthors on the article |
| Countries | Indicates the number of countries of the coauthors |
| Multi | Indicates whether the Mexican author(s) collaborates with two or more countries |
| BI | Indicates whether the Mexican author(s) collaborates with other country |
| Dom | Indicates coauthorship is among only Mexican institutions |
| Solo | Indicates if the document as written by a solo author |
| USA | Indicates if the document had a collaboration with at least one author from the USA |
| **Mexican Author Institution** | |
| UNAM | Indicates whether the institution of affiliation of the author(s) is the National Autonomous University of Mexico (UNAM) |
| IPN | Indicates whether the institution of affiliation of the author(s) is the National Polytechnic Institute (IPN) |
| UAM | Indicates whether the institution of affiliation of the author(s) is the Autonomous Metropolitan University (UAM) |
| CINVESTAV | Indicates whether the institution of affiliation of the author(s) is the Center for Research and Advanced Studies (CINVESTAV) |
| IMP | Indicates whether the institution of affiliation of the author(s) is the Mexican Petroleum Institute (IMP) |
| UDG | Indicates whether the institution of affiliation of the author(s) is the University of Guanajuato (UGU) |
| UGU | Indicates whether the institution of affiliation of the author(s) is the University of Guadalajara (UDG) |
| Others | Indicates if the institution of affiliation of the author(s) is any other in the country |

Source: Own elaboration.

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
