# Peer review of "Determining the Characteristics of Papers That Garner the Most Significant Impact: A Deep Dive into Mexican Engineering Publications"

_publications, doi:10.3390/publications11040046_

Round 1

Reviewer 1 Report

The manuscript “The impact of knowledge creation of Mexican engineering” depicts the results of bibliometric analysis of the papers published by Mexican engineers. Using a sample of 15-years Web of Science-indexed articles, the authors investigated several types of collaboration, subject distribution, number of authors per paper, the impact in terms of citations, and language of papers trying to find the dependence between each of these variables. It was concluded that higher number of countries per paper is correlated to higher citations. At the same time the papers by single authors were also well-cited as compared to papers with domestic collaboration. The manuscript seems to provide interesting and useful results both for scientometricians and engineers. However, I have several recommendations for improving the paper.

General comments

1.     I don’t think that the language is an illustrative category in the analyzed sample to be included in the test. As almost all papers in the sample are published in English, studying sporadic articles in other languages could lead to bias, as the authors admitted themselves. Instead, it can be interesting to study such variables as reviews, journal ranks and uncitedness. Perhaps, those papers by solo-authors with high number of citations are review articles or those published in first or second quartile journals. It will be interesting also to know which subject areas demonstrate the highest share of uncited papers. Even without additional types of analyses, I think the language variable should be removed from the test. Although the authors indicate that non-English papers are cited weaker than English-language, it can be correct only in Web of Science database, but it can be different in some regional databases, such as Scielo.

2.     Deeper interpretation of the results can enrich the manuscript. For instance, I’m interested why 2004 year was the starting point of intensive growth of scholarly output in engineering in Mexico? Were there any events stipulated that growth? Another example is the above mentioned situation with solo-authors publishing highly-cited papers. Are there any assumptions explaining this phenomenon? One more example: why the number of co-authors per paper have been steadily increasing in engineering, as it is shown in Fig. 2? Is it due to more active collaboration or some scientific policy in Mexico?

3.     In the methodology section, it should be stated that the sum of papers per subject areas will surpass the total amount of papers due to overlapping. It is stated only in Table legends. Please, indicate the level of overlapping (it is indicated in Table 4).

4.     In p. 9 the authors listed variables related to the most productive Mexican universities. However, these data are not discussed afterwards.

5.     At the end of the Conclusion section, the authors plan their future work. However, I don’t think it is a good idea to give references related to purposed works as in fact these studies are not carried out yet.

Minor comments

Trends in Fig. 1 are too vague. I think the shadows should be removed behind the lines.

P. 7, line 229 – change round brackets with references with square brackets.

P. 9: please, check the writing of characters from the formula (Ci or C1?).

P. 13, line 368: “number of authors in included” – is included?

Author Response

Reviewer 1.

  1. Considering that almost all the papers in the sample have been published in English, the reviewer suggested removing language as a control variable in the regressions.
    • In this new version of the paper we have dropped the language variable. Our results do not change in sign and significance, and changes in the value of the coefficients are minimal.

  1. The reviewer suggested deeper interpretation of the results. In particular, why 2004 was a starting point of intensive growth, highly cited papers published by solo-authors, and the increasing number of coauthors over time.
    • Related to why 2004 was a starting point of intensive growth. The analysis of the data suggests that this is the result of the creation of Scopus that year. According to the Science and Technology Indicators produced by the National Science Foundation (https://ncses.nsf.gov/pubs/nsb20214/data), the average rate of global publication output in engineering from 1997 to 2003 was 0.05%, and in 2004 and 2005 was 18% and 23%, respectively. We have added a footnote on page 4.

  • In the case of the solo authored papers, we have done a deeper analysis and the data suggest that the variance in citations is larger in solo-authored papers than in domestic collaboration, and the number of publications is much smaller. Only 36 (out of 405) do not have any citations and there is one with the highest number of citations is 515.

  • The growing number of coauthors in Mexico is similar to what other studies have found (Fortunato et al., 2018 and Thelwall & Maflahi, 2022). Page 6 -208.

  1. As suggested by the reviewer we have stated in the methodology section that the sum of paper per subject areas will surpass the total number of papers due to overlapping. (Pag 5, 188-190).
  2. As suggested, we have included a paragraph saying that controlling for the most productive institutions does not change the results and this suggests that it is not a key driver of the impact. (pag. 13 440-442)
  3. Thanks for noting this. We have clarified that the other studies that have been done related to gender gap differences analyze other countries and not Mexico. (pag. 14 488-489).

All minor errors were amended.

Reviewer 2 Report

The text presents the impact of knowledge in Mexican engineering. Its objective is to analyze the evolution of Mexican publications in the field of Engineering and their citations through the Web of Science from 2004 to 2017. To do this, it explores the characteristics of publications that tend to have a greater impact, that is, a higher number of citations. As a result, it highlights the need for internationalization of publications, suggesting multiple collaborations, writing in English, and involving authors from the United States.

The methodology is appropriate and detailed, showing the methods and materials used to achieve the objectives.

The results are clear and meet the objective of the study, showing that: diversification of countries increases the impact of research; highly cited articles are those written in multilateral collaboration [three or more countries]; collaboration with at least one co-author from the United States increases the number of citations received, which is identified in all models; and finally, articles in other languages receive fewer citations than articles in English.

Therefore, I am in favor of its publication.

Author Response

We appreciate the comments of reviewer 2.

Reviewer 3 Report

This paper analyzes the evolution of publications and citations in engineering in Mexico from 2004 to 2017, exploring the characteristics of publications that tend to have the greatest impact. The study considers variables such as the type of collaboration, number of coauthors and countries, and language of the article. The paper highlights that scientific collaboration brings complementary knowledge and expertise to tackle the increasing complexity of problems. The study concludes that multi-author papers receive more citations than solo-authored research, and international collaboration is associated with the greatest impact.

Overall, the manuscript is well-written and easy to follow. I think that the results are clear and sufficiently discussed. I would add a figure (or expand figure 1 to start from 1970) to support the choice made to include only articles published between 2004 and 2017 (see paragraph starting at line 160).

Another suggestion to the authors is to make figures 1 and 3 appropriate for grey-scale printing. For example, use different line styles (e.g., dashed, dotted, solid) or markers (e.g., circles, squares, triangles) in addition to colors to differentiate between multiple lines or data points.

Last but not least, in this work, the authors consider that the citation count reflects the impact of a scientific publication. However, lately Semantic Scholar has introduced additional indicators that reflect different aspects of the impact of a scientific publication. A highly related survey and experimental evaluation on the different impact indicators and how they capture different aspects of impact can be found in the following link:https://www.computer.org/csdl/journal/tk/2021/04/08836082/1di9U01TRdK

As a future work, it would be nice to see a similar analysis providing insights using such indicators, apart from the citation count. 

- line 38: "are" instead of "is"

- line 78: no "-" needed in the title of the section

- line 143: it is "an" objective measure that...

- line 272: related to "the" type of collaboration

- Tables 8 and 9: "USA" instead of "Usa"

Author Response

As suggested by the reviewer, Figure 1 was modified and now it includes the evolution since 1970. In addition, figures 1 and 3 were modified for a better visualization. Finally, we included (pag. 14 490-491) that in future work we are going to perform a similar analysis using new impact indicators such as those proposed by Semantic Scholar.

All minor errors were amended.

Round 2

Reviewer 1 Report

In the revised version of the paper “The impact of knowledge creation of Mexican engineering”, the authors have properly addressed all comments from my review. I believe the paper will be appropriate for publication after addressing several additional minor comments listed below.

Minor comments

1. All figure legends should be indicated below the pictures, not above them.

2. Table 5 seems to be redundant in this version of the manuscript, as the authors removed analysis of the language of the papers.

3. In the abstract, the authors indicated only some of the obtained results. It seems that an interesting result concerning single-authored papers cited better than domestic-collaborated papers should also be included.

4. The term “collaboration” should be added to the keywords as the main concept in the paper.

5. P. 3, line 101: the abbreviation WoS is mentioned for the first time here and should be explained. On the contrary, the explanation should be removed in the line 135 (it is the second mention).

6. P. 5, fig. 1: legend “articles” is overlapped with the values at the left.

7. P. 5, fig. 1: please, indicate which citations is meant in the figure. As I understand, it is the total amount of citations in each year to complete group of papers of all previous years. It should be specified in the figure legend, as the authors wrote about 5-year citation window in the Methodology section.

8. P. 6, line 193: “others field” – it seems that “other fields” should be written.

9. P. 10, Table 6: “Cited five-year window.” – remove dot point.

10. P. 14, line 426: “Kanellos et al [40]..” – et al. [40].

Author Response

  1. All figure legends should be indicated below the pictures, not above them.
    • We have changed the titles of the figures. In this new version of the titles are below the figures.
  2. Table 5 seems to be redundant in this version of the manuscript, as the authors removed analysis of the language of the papers.
    • We have deleted table 5.
  3. In the abstract, the authors indicated only some of the obtained results. It seems that an interesting result concerning single-authored papers cited better than domestic-collaborated papers should also be included.
    • We have added a paragraph stressing the result.
  4. The term “collaboration” should be added to the keywords as the main concept in the paper.
    • The term collaboration has been added as a keyword.
  5. 3, line 101: the abbreviation WoS is mentioned for the first time here and should be explained. On the contrary, the explanation should be removed in the line 135 (it is the second mention).
    • We have amended this error.
  6. 5, fig. 1: legend “articles” is overlapped with the values at the left.
    • We have amended this error.
  7. 5, fig. 1: please, indicate which citations is meant in the figure. As I understand, it is the total amount of citations in each year to complete group of papers of all previous years. It should be specified in the figure legend, as the authors wrote about 5-year citation window in the Methodology section.
    • We have amended this error.
  8. 6, line 193: “others field” – it seems that “other fields” should be written.
    • We have ameded this error.
  9. 10, Table 6: “Cited five-year window.” – remove dot point.
    • We have amended this error.
  10. 14, line 426: “Kanellos et al [40]..” – et al. [40].
    • We have amended this error.